# Contactless Measurements of Carrier Concentrations in InGaAs Layers for Utilizing in InP-Based Quantum Cascade Lasers by Employing Optical Spectroscopy

**DOI:** 10.3390/ma13143109

**Published:** 2020-07-12

**Authors:** Marcin Kurka, Michał Rygała, Grzegorz Sęk, Piotr Gutowski, Kamil Pierściński, Marcin Motyka

**Affiliations:** 1Laboratory for Optical Spectroscopy of Nanostructures, Department of Experimental Physics, Wrocław University of Science and Technology, Wybrzeże Wyspiańskiego 27, 50-370 Wrocław, Poland; michal.daniel.rygala@gmail.com (M.R.); grzegorz.sek@pwr.edu.pl (G.S.); marcin.motyka@pwr.edu.pl (M.M.); 2Łukasiewicz Research Network–Institute of Electron Technology, Al. Lotników 32/48, 02-668 Warszawa, Poland; gutowski@ite.waw.pl (P.G.); pierscin@ite.waw.pl (K.P.)

**Keywords:** Berreman effect, quantum cascade lasers, gas sensing, carrier concentration, mid-infrared

## Abstract

The precise determination of carrier concentration in doped semiconductor materials and nanostructures is of high importance. Many parameters of an operational device are dependent on the proper carrier concentration or its distribution in both the active area as well as in the passive parts as the waveguide claddings. Determining those in a nondestructive manner is, on the one hand, demanded for the fabrication process efficiency, but on the other, challenging experimentally, especially for complex multilayer systems. Here, we present the results of carrier concentration determination in In_0.53_Ga_0.47_As layers, designed to be a material forming quantum cascade laser active areas, using a direct and contactless method utilizing the Berreman effect, and employing Fourier-transform infrared (FTIR) spectroscopy. The results allowed us to precisely determine the free carrier concentration versus changes in the nominal doping level and provide feedback regarding the technological process by indicating the temperature adjustment of the dopant source.

## 1. Introduction

Mid-infrared is a spectral region of ever increasing significance. Numerous applications include free-space communication, imaging, and gas sensing [1]. The latter itself covers a vast area, ranging from medical diagnosis, e.g., detecting cancer markers [2] by breath analysis, localizing toxic or explosive leaks at factories and waste disposal sites, in-situ industrial process control, up to remotely checking alcohol content in exhaled air inside vehicles [3,4]. Absorption spectra with strong characteristic lines for different gases in that range make it possible to unambiguously identify gas mixture composition, and the state-of-the-art optical sensing systems are able to detect them at ppb concentrations in sub-second temporal resolution using quantum cascade lasers (QCLs) [5,6,7,8] or interband cascade lasers (ICLs) [9,10]. Many properties of those lasers are determined by properly chosen carrier concentration in respective areas thereof. Especially, the active regions of these lasers contain n-type doping, the concentration of which has to be precisely controlled to ensure the operation of the device and its performance. Moreover, they must be cladded by a layer of semiconductors with proper refraction index, in order to make a waveguide for the generated radiation. This can be achieved by changing the charge carrier concentration via precisely controlling the amount of doping during growth. Verifying the concentration levels during post-processing is normally destructive, by using such techniques as Hall effect measurements or capacity-voltage measurements [11]. There are, however, several optical experiments which not only are contactless, but in general, do not require affecting the sample, which can be then further processed with already known characteristics. With no preprocessing requirements, and the possibility of measuring grown wafer during the process, it seems a very promising upgrade. In this paper, we show results of carrier concentration measurements using so the called “Berreman effect” [12]. Our purpose was to establish the carrier concentration of In_0.53_Ga0_.47_As layers and compare it with “nominal”, i.e., obtained by interpolation of Hall measurements of reference samples’ concentration, and to make improvements to the doping process. The material of the samples was purposed for active areas of QCLs for Long Wavelength Infrared (LWIR) range. The method itself allows to measure various kinds of layers with thicknesses below 10 nm [13], with reports that the effect takes places in ~1 nm thick region [14,15]. The concentrations measured can be as low as 10^17^ cm^−3^ while performing measurements in transmission mode [16], given sufficient sensitivity of the setup, which can be improved using modulation techniques such as fast differential reflectance [17,18], or photoreflectance in step-scan mode of the FTIR spectrometer [19,20]. The phenomenon can be observed as an enhancement of absorption of p-polarized radiation where the dielectric function reaches zero at the plasmon frequency. In heavily doped semiconductors, this can be ascribed to free carriers in surface plasmon polaritons, and its frequency is directly dependent on the carrier concentration. Therefore, it is possible to determine the concentration of free carrier in a sample by measuring the changes in the absorption of polarized light.

## 2. Materials and Methods

In this paper, we show the results of reflectance measurements of the Berreman effect of In_0,53_Ga_0,47_As layers samples with different nominal carrier concentrations. These are 1-µm thick Si-doped In_0.53_Ga_0.47_As layers lattice-matched to the InP substrate, grown by solid source molecular beam epitaxy (MBE) with a Riber Compact 21T reactor [21]. Different temperatures of the Si-source were used in order to vary the doping concentrations. The nominal carrier concentrations in the investigated structures were calculated by interpolation of Hall measurement curves measured with a Bio-Rad HL5500 system (Bio-rad, Hercules, CA, USA), and are shown in Table 1.

The optical measurements were performed in Bruker Vertex 80 FTIR spectrometer (Bruker, Billerica, MA, USA) with additional custom-designed evacuated external chamber to provide an oblique angle of 45 degree for the incident light [17,18] (which is necessary to observe the Berreman effect). As the light source, a glowbar was used, whereas the detector was a liquid-Nitrogen-cooled Mercury Cadmium Telluride photodiode.

## 3. Results

Figure 1 shows reflectance spectra for all samples measured at 300 K for two orthogonal linear polarizations of the incident light. A distinct absorption dip can clearly be seen for TM polarization (red curves), which corresponds to a case where there is an electric field component perpendicular to the sample surface. Black curves denote spectra obtained for s polarized light (TE) where there is no corresponding minimum observed, which is consistent with theory. Also visible in the red curves of the TM polarization is how the Berreman minima shift towards lower wavenumber (longer wavelength) with decreasing doping and finally even disappear for sample D due to the cut-off wavelength of our setup (detector limit at 600 cm^−1^).

Figure 2 shows intensity normalized reflectance spectra for the p-polarized light of three samples A, B and C. The absorption minima can be ascribed to plasma frequency *ω_p_*_,_ of free electrons in the layers. We can see that *ω_p_* shifts to lower energies with carrier concentration decreasing according to Equation (1).
(1)$$\omega_{p}^{2}=\frac{ne^{2}}{\varepsilon_{0}\varepsilon_{\infty }m^{*}}$$
where *ε*_∞_ is the material dielectric constant equal to 11.64 and *ε*_0_ is the permittivity of free space. The value of *m** is 0.0453 of electron rest mass.

Figure 3 shows a comparison of the dependence of the obtained nominal carrier concentration versus plasma frequencies (black squares) obtained for InGaAs alloy with similar composition by Charache et al. [22]. This allowed us to get a useful function after exponential fitting (red line). A similar procedure has been proven to be successful when applied for the analysis of Berreman minima in the case of carrier concentration determination in InAs layers [23], using data from Hinkey et al. [24]. By open squares we marked nominal concentration values on the fitted line at the respective wavenumbers. Open solid points denote the plasmon frequencies measured by reflectivity measurements (given in wavenumbers). By placing these frequencies on the fitted curve, we were able to determine the actual electron concentrations of 1.3 × 10^19^ cm^−3^, 7.0 × 10^18^ cm^−3^, and 4.0 × 10^18^ cm^−3^ for samples A, B, and C, respectively. Furthermore, we can see that the revealed differences between the nominal and the measured concentrations, despite not being large, show for all cases the measured values to be smaller than the nominal ones. This is just the first approximation approach, which can be used as an attempt for growth parameters verification in order to establish a better match between nominal and achieved concentrations in the layers.

A more sophisticated and precise method requires including the effective mass dependence on the carrier concentration in the calculation of the plasma frequencies as a function of concentration, since it cannot be neglected for concentrations above 10^18^ cm^−3^, as in ref. [25]. Due to non-parabolic behavior of energy dispersion far from Γ point of the Brillouin zone, a correction of carrier effective mass must be taken into account, which can be expressed by Equation (2) taken from ref. [26].
(2)$$m^{*}\left(n\right)=m_{e}{\left[1+\frac{4P^{2}}{3E_{g}}{\left(1+\frac{8P^{2}\hbar^{2}{\left(3\pi^{2}n\right)}^{\frac{2}{3}}}{3m_{e}E_{g}^{2}}\right)}^{\frac{-1}{2}}\right]}^{-1},$$
where *m_e_* is the electron rest mass and *P*^2^ is the momentum matrix element of coupling between valence and conduction bands, calculated to be 15.4 eV.

Figure 4a shows the calculated *m*/m_e_* ratio as a function of electron concentration. As we can see, the role of correction is more important for larger concentrations in the range of nominal concentrations considered within this paper (indicated by the red dotted square), where the relative changes of the effective mass up to around 30% can occur. Figure 4b shows calculated plasma frequencies assuming the mass correction (black curve) together with the nominal values (black squares) and those determined here (red circles). The obtained values are still slightly lower than nominal, however closer to the nominal values when compared to those obtained in the first, simpler approach. The respective summary is shown in Table 2. The resolution of our experimental setup was 2 cm^−1^, while the full widths at half maximum (FWHM) of measured spectra were ~50 cm^−1^. This allowed us to estimate uncertainty of the carrier concentration at ±2%, which is not enough to explain the difference between nominal concentration.

## 4. Discussion

Figure 5 shows the nominal electron concentrations for all investigated samples versus the applied Si source temperature together with a fit (black curve) which shows this dependence in a broader range of concentrations and temperatures. Red open circles denote determined (by second more accurate approach) electron concentrations for samples A–C together with respective fit (red curve). The comparison of the nominal dependency and the corrected experimental results shows that the respective adjustments become important for larger concentrations (higher source temperatures).

Therefore, in a range of concentrations presented in this paper, a slight temperature increase seems to be appropriate. For instance, to obtain a nominal concentration for sample A, Si source temperature should be increased by ~15 K from 1300 °C to 1315 °C to offset the observed differences. On the other hand, in the range of typical doping (~10^16^–10^17^ cm^−3^) of the lasing active areas of quantum cascade lasers, little temperature compensation would be required.

## 5. Conclusions

In this paper, we have demonstrated an optical method used for electron concentration determination in the calibration of InGaAs layers designed for waveguides of InP-based quantum cascade lasers. This method requires neither contacts nor an external magnetic field as are typically required in direct-contact methods such as Hall and C-V measurements. Moreover, the method can be applied to various types of structures, made of different kinds of materials and grown in different techniques such as metal-organic chemical vapor deposition, as well as different techniques of doping. It is important to note that 15 K is a significant difference from the point of view of manufacturing technology. There are several possible causes that would explain reported discrepancies between Hall effect method and the Berreman effect method, such as the degree of ionization of the dopant atoms in the sample, or gauging the magnetic field of the magnet, while using Hall setup. Further establishing causes of shown differences is crucial to optimize the manufacturing process and therefore the quality of ready devices due to the feasibility of the optical method, which makes it a perfect candidate to supersede other methods. By determination of the so called “Berreman effect minima” in the reflectance spectra and the derived carrier concentration versus plasma frequency, we were able to verify the nominal concentration, and finally establish Si source temperature versus concentration dependence.

## Figures and Tables

**Figure 1 materials-13-03109-f001:**
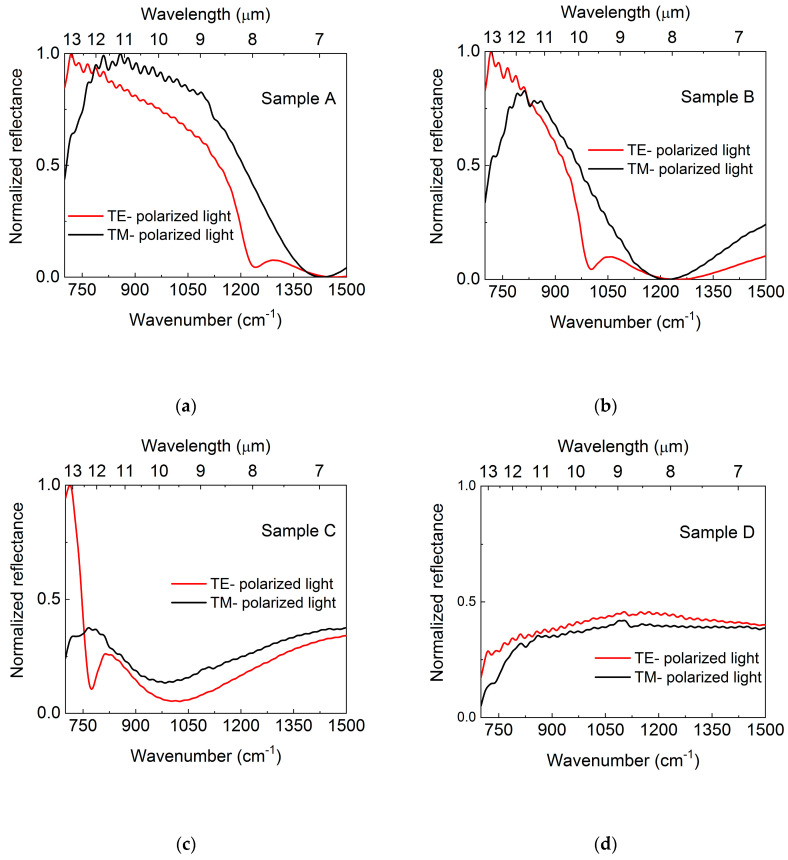
The reflectance spectra for two orthogonal polarizations of the probing light denoted TE (red curves) and TM (black curves): (**a**) 1.9 × 10^19^ cm^−3^; (**b**) 9.9 × 10^18^ cm^−3^; (**c**) 5.2 × 10^18^ cm^−3^; (**d**) 1.1 × 10^18^ cm^−3^.

**Figure 2 materials-13-03109-f002:**
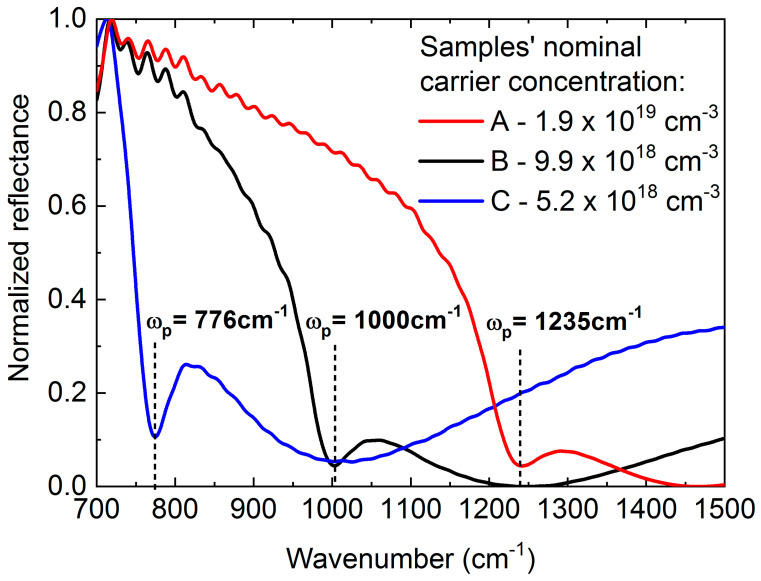
Normalized p-polarized reflectance spectra for Sample A (red curve), Sample B (black curve), and Sample C (blue curve) together with given characteristic reflectance minima at plasma frequencies and indication of carrier nominal concentration.

**Figure 3 materials-13-03109-f003:**
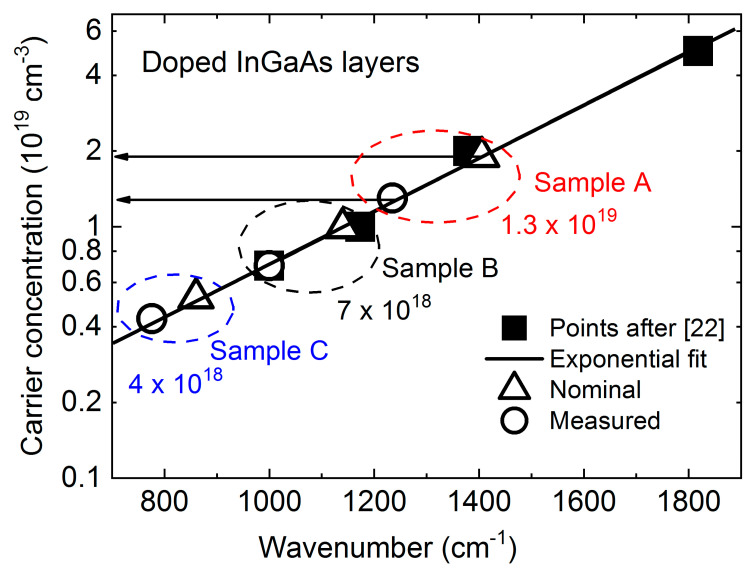
Carrier concentration in a function of Plasma frequency. Black squares denotes points after ref. [22] together with respective fit (red line). Triangular points represents nominal concentrations and open circles those established.

**Figure 4 materials-13-03109-f004:**
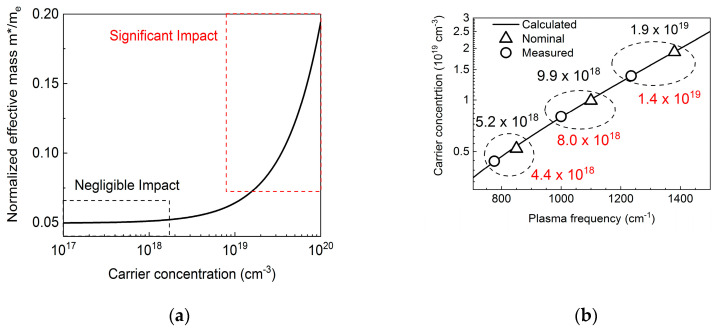
(**a**) Electron mass changes as a function of their concentration after Equation (2); (**b**) Plasma frequencies as a function of electron concentration. Black curve represents calculated Plasma frequencies assuming mass correction shown in panel (**a**). Open triangles depicts nominal values and open circles those measured.

**Figure 5 materials-13-03109-f005:**
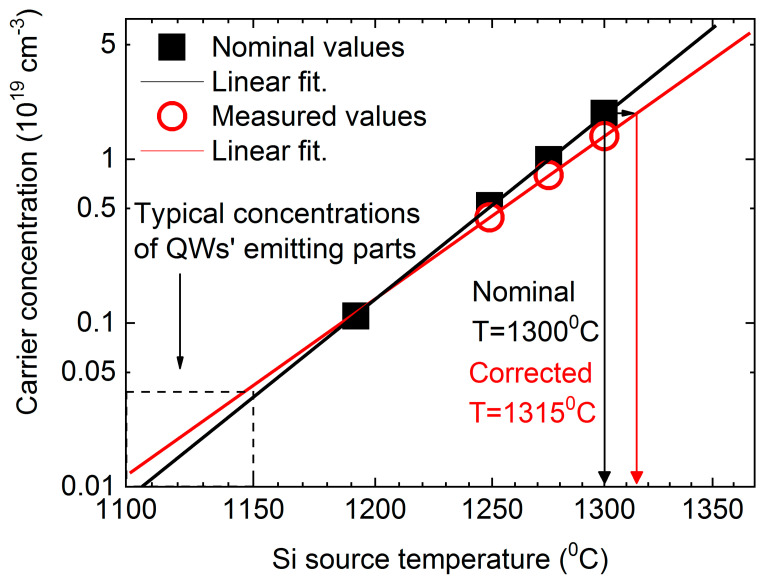
Nominal carrier concentrations (black squares) as a function of applied Si source temperature together with fit (black curve). Open red circles denotes corrected (in the way of second approach) values of the carrier concentration in a function of the applied Si source temperature together with respective fit (red curve).

**Table 1 materials-13-03109-t001:** The samples’ description.

Sample	Hall-Measured Carrier Concentration (cm^−3^)	Si-Source Nominal Temperature (°C)
(A) C766	1.9 × 10^19^	1300
(B) C764	9.9 × 10^18^	1275
(C) C763	5.2 × 10^18^	1250
(D) C759	1.1 × 10^18^	1190

**Table 2 materials-13-03109-t002:** The samples’ description and growth protocol.

Sample	Nominal Carrier Concentration	First Approach	Second Approach
(A) C766	1.9 × 10^19^	1.3 × 10^19^	1.38 × 10^19^
(B) C764	9.9 × 10^18^	7.0 × 10^18^	8.0 × 10^18^
(C) C763	5.2 × 10^18^	4.0 × 10^18^	4.4 × 10^18^
(D) C759	1.1 × 10^18^	N/A	N/A

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
