# Peer review of "Contactless Measurements of Carrier Concentrations in InGaAs Layers for Utilizing in InP-Based Quantum Cascade Lasers by Employing Optical Spectroscopy"

_materials, 2020, doi:10.3390/ma13143109_

Round 1

Reviewer 1 Report

At first glance, I got the impression that the paper by Marcin Kurka et al., has been fast and carelessly written.

Just to mention a few “technical errors” detected by this reviewer,

  1. Equation (1) corresponds to the square of the plasma frequency, not to the plasma frequency itself.
  2. The caption of figure 2 is wrong; the color code does not correspond with the labels in the inset of the same figure.
  3. The caption of figure 3 is wrong; REF 15 should read REF 16.
  4. If figure 4(b) is meant to be an improved version of figure 3 (taking into account the carrier concentration dependence of the electron effective mass), why the authors don’t plot fig. 4(b) with the same X and Y axes as in figure 3, for direct comparison?
  5. The two tables are poorly edited.
  6. In line 138, "sample C" should read "sample A".

At the end of the day, the leitmotif of the present paper is to demonstrate that the carrier concentration of In0.53Ga0.47As layers, a material relevant for quantum cascade lasers, can be extracted by IR reflectance measurements of the Berreman effect.

This has been already demonstrated on the same material in the excellent paper from J. Ibáñez, et al. PHYSICAL REVIEW B 69, 075314 (2004). Moreover, this paper covers the low carrier concentration region ~10exp17, relevant for quantum cascade lasers.

Moreover, when compared to the carrier concentration extracted from Hall measurements (what the authors label as “nominal values”), there is a small discrepancy, and the values extracted by IR reflectometry are systematically lower. It seems to me that the authors assume that the correct values are the ones extracted from optical measurements, without any further justification.

For all the reasons explained above, in my opinion the present paper cannot be accepted to be published in Materials Journal.

Reviewer 2 Report

Review of “Contactless measurements of…” for publication in MDPI Materials Journal.

In this paper, the authors do a detailed comparison of Hall-effect measurements for concentration for four, doped samples of lattice matched InGaAs with reflectivity measurements derived from the Berreman effect. Eventual results were within 10-20% of Hall effect measurements, after including for corrections in effective mass.

Development of contactless techniques for carrier measurement is an important topic and should definitely be pursued.The paper itself is thorough and well presented. There has been some literature on this topic already presented, including

Hinkey, Robert T., Zhaobing Tian, Rui Q. Yang, Tetsuya D. Mishima, and Michael B. Santos. "Reflectance spectrum of plasmon waveguide interband cascade lasers and observation of the Berreman effect." Journal of Applied Physics 110, no. 4 (2011): 043113.

(which should probably be cited) and the authors own,

Dyksik, M., M. Motyka, G. SÄ™k, J. Misiewicz, M. Dallner, Sven Hoefling, and M. Kamp. "Influence of carrier concentration on properties of InAs waveguide layers in interband cascade laser structures." Journal of Applied Physics 120, no. 4 (2016): 043104.

This paper represents an incremental addition to the same topic.

A few comments on the structure of the paper; essentially the paper is about methods for contactless carrier measurement, but the introduction talks most about applications for QCL. That should be revised. The information about the Si source temperature for reactor sample is irrelevant and could be removed : the paper should be valid for MOCVD grown layers as well, and systems often have minor variations. The reviewer would also like to see a discussion of the applicability of this technique; how thick does the layer have to be? What range of doping can it be ultimately applied to?

Reviewer 3 Report

The manuscript reports measurements of carrier concentrations in InGaAs layers with a contactless method based on the Berreman effect. This method is attractive because of its simplicity. However, in order to evaluate its usefulness for quantum cascade laser (QCL) fabrication, several important aspects listed below should be discussed. I recommend publication after revisions to discuss these points.

1. What is the domain of application of the reported method, i.e. the range of carrier concentrations that can be measured? With your FITR detection limit of 600 cm-1 you can only measure concentrations > 1E18 cm-3. The typical carrier concentrations in QCLs are significantly lower than that, of the order of 1E16-1E17 cm-3. Only the uppermost layers of the top cladding have carrier concentrations in the 1E18-1E19 cm-3 range and the precision of the precision of the doping calibration in these layers is not critical.

2. The manuscript emphasizes the precision of the method but no error bars are given on any of the measurements. It is important to estimate the absolute accuracy and the reproducibility of the method to allow comparison with other techniques. A direct comparison of measurements obtained with different methods would also be useful. 

3. What are the requirements on the samples to apply this method? It is non-destructive but if it cannot be applied to real QCL wafers and requires to grow dedicated wafers, then it is not really an advantage.

I noticed several mistakes in the text:

Line 62: "vacuumed external chamber". You probably mean evacuated. Vacuumed means cleaned with a vacuum cleaner.

Line 80: "characteristic absorption minima". These are reflectance minima.

Equation 1: the left-hand side should be wp^2, not wp.

Line 89: "after linear fitting". I believe this is an exponential fit because it is straight line on a log plot.

Fig. 3: "points after [15]". Charache et al. is Ref. [16], not [15].

Reviewer 4 Report

The manuscript is relatively well written, however, grammatical corrections is still needed, use of too many generalized adverb terms such as simply, especially, so-called, etc. should be minimal. Overall figure quality is good except Fig 1&4. Authors should use different symbols and increase the size of the fonts: legend, labels, etc. What are these A, B, C, D in the caption is not clear. There is no point of repeating the same name both in Fig 1 and in the caption, instead, authors should give the actual name of the Samples in the Captions instead of just A, B, C, D. Do these samples have any physical names? Define these symbols in the caption to assist the readers. 

Reviewer 5 Report

In this manuscript, the authors present their study on the contactless optical measurements of carrier concentrations in InGaAs layers utilizing the Berreman effect. The carrier concentration was estimated through the plasma frequency according to the Drude model. The obtained carrier concentration moderately agreed with that measured by the Hall measurement. I think that the results are interesting and demonstrate the potential of the nondestructive carrier concentration measurements in the fabrication process. However, I have several comments on the discussion in the manuscript. Before this manuscript can be accepted for publication, the authors need to revise their manuscript according to the following comments.

Comments:

1) As the authors mentioned in the conclusion, the source temperature difference of 15 K cannot be ignored. What is the possible origin of the discrepancy in the source temperature between the Hall and Berreman effect methods?

2) ωp in Eq. 1 should be ωp2. The carrier concentration was estimated through the plasma frequency according to Eq. 1 in this work. I guess that quantum-mechanical calculation is required for estimating the carrier concentration more precisely. For example, Paskov calculated the optical constants in narrow-band-gap semiconductors based on numerical Kramers–Kronig analysis of the carrier-related imaginary part of the dielectric function [P. P. Paskov, J. Appl. Phys. 81, 1890 (1997).].

3) It seems to me that the purpose of the work is not clearly written in the introduction section.

Minor comments:

4) The sample name in Table 1 should be written in one line for the samples A and D.

5) The legends in Fig. 1 are written oppositely; TE and TM polarizations should be black and red, respectively.

6) Black squares in Fig. 3 would be referred from Ref. 16.

Round 2

Reviewer 1 Report

The authors have properly addressed most of my previous comments on the manuscript. However, I think they should carefully re-read it and fix any possible errors and complete the information detailed below, before the paper can be accepted for publication

Things to be revised:

  1. Mistake in caption of Figure 1, (d) should read 1.1exp18.
  2. The authors should mention the electron effective mass value and the material dielectric constant value used in figure 3, according to equation (1).
  3. As seen in Figure 4(a) the correction for the effective mass value is getting more important for higher values of carrier concentration. According to this, the difference in the carrier concentration extracted from method_1 or method_2 should increase increasing carrier concentration (from sample (c) to sample (a)) and is not de case, see table 2.
  4. If graph 3(b) and graph 4 display the same physical variables, I don’t understand why the axis titles(X and Y) are not the same in both figures?.

Reviewer 3 Report

The authors addressed all the concerns that I had with the first version. I recommend to publish the manuscript in its present form.

Author Response

The authors would like to thank the reviewer.